# Establish TiNb_2_O_7_@C as Fast-Charging Anode for Lithium-Ion Batteries

**DOI:** 10.3390/ma16010333

**Published:** 2022-12-29

**Authors:** Shuya Gong, Yue Wang, Meng Li, Yuehua Wen, Bin Xu, Hong Wang, Jingyi Qiu, Bin Li

**Affiliations:** 1State Key Laboratory of Organic-Inorganic Composites, Beijing Key Laboratory of Electrochemical Process and Technology for Materials, Beijing University of Chemical Technology, Beijing 100029, China; 2Research Institute of Chemical Defense, AMS, Beijing 100191, China; 3School of Materials Science and Engineering, University of Science and Technology Beijing, Beijing 100083, China

**Keywords:** TiNb_2_O_7_, carbon-coating, fast-charging, anode materials

## Abstract

Intercalation-type metal oxides are promising active anode materials for the fabrication of safer rechargeable lithium-ion batteries, as they are capable of minimizing or even eliminating Li plating at low voltages. Due to the excellent cycle performance, high specific capacity and appropriate working potential, TiNb_2_O_7_ (TNO) is considered to be the candidate of anode materials. Despite a lot of beneficial characteristics, the slow electrochemical kinetics of the TNO-based anodes limits their wide use. In this paper, TiNb_2_O_7_@C was prepared by using the self-polymerization coating characteristics of dopamine to enhance the rate-performance and cycling stability. The TNO@C-2 particles present ideal rate performance with the discharge capacity of 295.6 mA h g^−1^ at 0.1 C. Moreover, the TNO@C-2 anode materials exhibit initial discharge capacity of 177.4 mA h g^−1^, providing 91% of capacity retention after 400 cycles at 10 C. The outstanding electrochemical performance can be contributed to the carbon layer, which builds fast lithium ion paths, enhancing the electrical conductivity of TNO. All these results confirm that TNO@C is a valid methodology to enhance rate-performance and cycling stability and is a new way to provide reliable and quickly rechargeable energy storage resources.

## 1. Introduction

The global development of the need for green and sustainable energy has become urgent, and lithium-ion batteries are employed in practically all items, from hybrid and pure power automobiles to personal portable electronic gadgets [1]. As a result, the advancement of lithium-ion battery technology has become a key priority, especially since the performance of present batteries is primarily determined by the materials [2,3]. With the advantage of inexpensive, structural stability and high specific capacity (372 mA h g^−^^1^), graphite materials are commonly employed in LIBS, but the production of SEI films during the first cycle causes a partial loss of capacity and is prone to the precipitation of lithium dendrites, which leads to internal short circuits in the battery, posing a severe safety threat [4]. Due to the decreased risk of Li plating at low voltages, metal oxides with intercalation-type lithium storage mechanisms are being considered as an alternative anode material for safe LIBs. Niobium-titanium oxides are the intercalation-type anode materials which could provide fast charging capability for lithium-ion batteries and have excellent structural stability during lithiation/delithiation [5].

TiNb_2_O_7_ (TNO) possesses a layered monoclinic framework and can accommodate various ions [6]. Thus, it is often considered as an active LIB anode material as it offers high capacity, stable cycle performance, and safety [7,8,9]. Recently, various methods have been used to synthesize TNO anode materials and the electrochemical properties of the various TNO anodes have been supplemented in the Table 1. This Wadsley–Roth phase TNO exhibits a theoretical specific capacity of 387 mA h g^−^^1^ due to the TNO structural unit containing Ti^4+^/Ti^3+^, Nb^5+^/Nb^4+^and Nb^4+^/Nb^3+^ redox couples [10]. In addition, its interconnect tunnel construction along the b axis supplies Li^+^ pathways, ensuring the high-rate performance of the TNO-containing LIB [11]. Furthermore, owing to the lithiation voltage (1.6 V vs. Li^+^/Li), Li dendrite and SEI film production are avoided during the charging/discharging process, ensuring the safety of the battery [12]. Despite its excellent electrochemical performance, its slow ionic diffusion rate and low electronic conductivity severely hamper its electrochemical performance [13]. To address this issue, a variety of techniques have been used, including nanocrystallization, carbon coating, and doping [10,14,15,16,17]. Liu et al. [18] prepared 3D-porous TiNb_2_O_7_/CNT-KB composite microspheres by spray-drying and solvothermal method, which presents a high reversible charge specific capacity of 151.1 m Ah g^−^^1^ at 20 C. It retains 145.4 mA h g^−^^1^ after 1000 cycles at a high current rate of 5 C, with a capacity retention of 65.9%. Hsiao et al. [19] demonstrated that doping with W^6+^ could enhance TNO electrochemical performance and its reversible capacity reached 156.2 mA h g^−^^1^, even at a rate of 20 C. Shang et al. [20] made TNO/CNTs by ultrasonic dispersion and a facile solvothermal method. A high-reversible capacity achieved 261.1 mA h g^−^^1^ at 50 mA g^−^^1^. Furthermore, a noteworthy rate capability maintains 110 mA h g^−^^1^ at 500 mA g^−^^1^ with over 1000 cycles.

In this work, we prepared TNO@C by using the self-polymerization coating characteristics of dopamine to enhance the rate-performance and cycling stability. Additionally, the electrode’s electrical conductivity is significantly increased by the carbon layer, providing more channels for lithium ion transport [23]. With these benefits, the electrochemical performance of the TNO@C composite particles exhibits significant superiority compared to pristine TNO. The produced TNO@C-2 composite particles exhibit a significant charge/discharge capacity of 293.3/295.6 mA h g^−^^1^ at 0.1 C and 164.1 mA h g^−^^1^ after 400 long cycles at a high current density of 10 C, demonstrating the perfect reversible capacity and long cycling life. This work provides a method for the preparation of anode materials with fast charging and stable electrochemical performance.

## 2. Materials and Methods

### 2.1. Synthesis of TNO

TNO was made by a one-step solid state reaction using TiO_2_ from Macklin (Shanghai, China), with a purity of 99.8%, and Nb_2_O_5_ Macklin (Shanghai, China), with a purity of 99.9% mixed in a 1:1 molar ratio as precursors and then heated in a furnace (GSL-1400) at 1200 °C for 20 h under constant N_2_ flow. After the annealing, the powders were allowed to cool naturally with the furnace.

### 2.2. Synthesis of TNO@C

The samples which were prepared from the molar ratio of TNO and dopamine of 2:1, 1:1, 1:2 are denoted as TNO@C-1, TNO@C-2, TNO@C-3. TNO particles and dopamine were added to a tris-buffered aqueous solution (prepared by dissolving trimethylolaminomethane in deionized water) at Ph = 8.5 at 2:1, 1:1, 1:2 and reacted with magnetic stirring for 24 h to obtain polydopamine (PDA)-coated TNO (PDA@TNO). After washing with deionized water and drying, the PDA@TNO particles were treated in a furnace (GSL-1400) at 600 °C for 6 h in the flowing N_2_ to obtain TNO@C (Figure 1).

### 2.3. Material Characterization

The resulting materials were analyzed by powder X-ray diffraction (XRD) performed by a Rigaku-D/MAX-rA instrument (Tokyo, Japan) equipped with a Cu-Ka radiation source under 40 mA and 40 kV. The XRD patterns were collected over the 2θ range of 5–65° with a step size of 0.02°, and the scanning rate was 2° min^−1^. All materials were also characterized by scanning and transmission electron microscopies (SEM and TEM, respectively) performed by the Hitachi S-4800 and JEOL 2010 instruments (Tokyo, Japan), respectively. Raman spectroscopy was performed by Raman spectrometer (HORIBA Scientific LabRAM HR Evolution, France) to test the characteristic peaks of carbon (D band and G band), at room temperature. A four-point probe approach was used to determine the electrical conductivity of all the samples.

### 2.4. Electrochemical Measurements

For electrochemical measurements, CR2025 coin cells were prepared. The electrodes were prepared by mixing reactive TNO, acetylene black and PVDF in N-methylpyrrolidone at a ratio of 7:2:1. The slurry was spread uniformly on a Cu current collector and dried to obtain the electrode. The loading mass of the active material in the prepared electrodes was controlled in the range of 1.0~1.5 mg cm^−^^2^. The electrode thickness was about 25–30 μm and the Cu foil is 12 μm. Lithium mental (China Energy Lithium Co., Ltd., diameter × thickness: ϕ16 × 2 mm, >99.9%) was used as the counter electrode, and the separator consisted of a polypropylene membrane (Celgard H1609, thickness:16μm) that had been moistened with about 100 µL electrolyte (1 M LiPF_6_ and 1:1 volume combination of ethylene carbonate (EC) and Diethyl carbonate (DEC)).

Galvanostatic charge/discharge tests were performed on a LAND CT2001A (Wuhan, China), where 1 C = 388 mA h g^−^^1^, and cyclic voltammetry and electrochemical impedance spectroscopy data were measured on a CHI 600E electrochemical workstation (Shanghai, China). The frequency range of EIS was tested between 10^6^ and 10^−^^2^ Hz. The cyclic voltammetry (CV) scan rate was varied from 0.1 to 1 mV s^−^^1^ throughout a potential range of 1.0–3.0 V (vs. Li/Li^+^).

## 3. Results and Discussion

X-ray diffraction (XRD) was used to identify the detailed crystal structures of TNO@C-1, TNO@C-2, TNO@C-3, and pure TNO. Without any obvious impurity phases (TiO_2_ and Nb_2_O_5_), all of the sharp diffraction peaks can be assigned to the monoclinic TNO (00-434-3509). The appearance of the peaks at 2θ of 23.9° and 26.3° corresponded to the TNO crystal planes (−110) and (402) [24]. Furthermore, the samples of TNO@C exhibit the same diffraction peaks as TNO, proving that the TNO particles were not reduced (Figure 2a). In order to prove the carbon element exists inside TNO@C, Figure 2b shows the Raman date of TNO and TNO@C, which is located at 1328 and 1594 cm^−1^, corresponding to the characteristic D and G peaks. The peaks at 1002 and 889 cm^−1^ may also be associated with NbO_6_ octahedra with edge-shared and corner-shared corners. Both the peaks at 647 cm^−1^ and 539 cm^−1^ may be connected to the preferentially occupied edge-shared TiO_6_ octahedra [23,24].

SEM images were performed to better observe the microstructure and morphology of TNO and TNO@C. As exhibited in Figure 3a,b, the pristine TNO particles have a smooth surface with an irregular microstructure and an average particle diameter of approximately 1–2 μm. Carbon is encapsulated on the TNO surface, as evidenced by SEM of the TNO@C composites (Figure 2c–h). Figure 3i depicts the EDS elemental mapping of TNO@C, which shows the presence of Ti, Nb, O, and C elements in the material and a similar distribution of C elements to the others, suggesting that TNO@C samples were successfully prepared.

The surface of the pristine TNO particles is smooth and in the TEM image a d-spacing of 0.24 nm can be assigned to the (−512) crystal plane in TiNb_2_O_7_ (#39-1407) (Figure 4a,b). As can be seen in Figure 4c,d, the TNO@C-1 carbon layer is thin and partially uncoated due to the little amount of polydopamine. In addition, the TEM pictures of TNO@C-2 further reveal that carbon evenly covers the particle surface with a thickness of about 50 nm (Figure 4e,f). TNO@C-3 is also unevenly coated, with the thinnest part being 75 nm and the thickest reaching 200 nm (Figure 4g,h). The amount of carbon in the TNO@C samples was tested by organic elemental analysis and the results showed that the amounts of carbon in the samples were about 4.7, 5.2 and 6.4%. Furthermore, a superior electrochemical performance is also obtained by the carbon layer’s long-range electrical conductivity and effective enhancement of the electronic conductivity of the TNO particles [25].

Cyclic voltammetry (CV) measurements were performed on TNO and TNO@C electrodes to explore the electrochemical mechanism. The tests were done and the accompanying CV curves were recorded and compared. (Figure 5a) The position of the CV peak at the electrode cathode in the first cycle differed from that in the following two cycles, owing to irreversible lithiation in the first cycle. The TNO@C samples have smaller equivalent peak shifts than the pure TNO, indicating that carbon coating reduces irreversible lithiation. Furthermore, the CV curves displayed a pair of strong electrode cathode/anode peaks at 1.6/1.68 V, which may be attributed to the Nb^4+^/ Nb^5+^ redox couple, and a pair of shoulder peaks at 1.76/1.8 V, that can be related to the Ti^3+^/Ti^4+^ redox reaction. Meanwhile, the Nb^3+^/ Nb^4+^ redox process is responsible for the wide bump from 1.0 V to 1.4 V [26]. Apart from the pure TNO, the CV curves of TNO@C samples were almost fully overlapping. Thus, the carbon coating could improve cycle stability [27].

As shown in Figure 5b, the samples of TNO-C provide better discharge capacities (273.5, 295.6, and 286.6 mA h g^−1^ for TNO@C-1, TNO@C-2, and TNO@C-3) with coulombic efficiency of 96.7%, 99.2%, and 96.8%. Meanwhile, TNO had an initial discharge capacity of 266.4 mA h g^−1^ and a coulombic efficiency of 98.4%. As a result, the carbon coating improves its discharge capacity. The proper amount of carbon improves its coulombic efficiency; on the other hand, an excessive or insufficient amount of carbon might cause side effects. TNO delivered 245.7, 232.3, 218.8, 197.2, 160.5, and 130.4 mA g h^−1^ at 0.1, 0.5, 1, 2, 5, and 10 C, respectively (Figure 5c,d). At the same rate, TNO@C-2 has a larger capacity than TNO@C-1 and TNO@C-3, with 294.6, 265.3, 241.7, 221.5, 190.5, and 164.4 mA h g^−1^, respectively. Moreover, TNO@C-2 also exhibits good reversibility, when the current density is reduced to 0.1 C, the capacity is 290.4 mA h g^−1^. To further demonstrate the effect of carbon coating on electrochemical performance, cycling tests were carried out on TNO and TNO@C. The TNO@C-2 anode showed that the discharge capacity of 248.7 mA h g^−1^ after 100 cycles at 1 C (Figure 5e). Furthermore, TNO@C-2 exhibited the best electrochemical performance with the highest capacity and excellent cycling stability. Even at 10 C, TNO@C-2 displayed an initial discharge capacity of 177.4 mA h g^−1^ and the capacity only reduced by roughly 9% after 400 cycles, the capacity retention of TNO@C-2 is higher than TNO@C-1 (78.8%) and TNO@C-3 (79.6%), while, TNO only provided the discharge capacity of 101.3 mA h g^−1^ with 65 % retention after 400 cycles at 10 C. (Figure 5f). Carbon coatings increase electrical conductivity and reduce irreversible processes, which result in significant improvements in rate performance and cycling stability in TNO@C composites.

In order to obtain insight into the improved electrochemical characteristics of the TNO@C composite, EIS analyses were performed on both TNO and TNO@C-2 samples. Figure 6a depicts the appropriate Nyquist plots and equivalent circuit, which consists of a semicircle in the high and medium frequency ranges (10^6^–10^4^, 10^4^–10^1^)) and a sloping line in the low frequency region (10^1^–10^−2^). The semicircle in the high frequency range is due to the impact of lithium ion insertion on the particle surface (R_SEI_); the other semicircle in the medium frequency range is due to electron transfer, which is represented in the equivalent circuit by R_ct_, the slope in the low frequency region corresponds to Warburg resistance (W_o_), reflecting diffusion impedance [28]. For the TNO sample, the fitted R_SEI_ and R_ct_ values are 4.5 and 50.9 Ω, respectively. On the other hand, the TNO@C-2 are just 2.6 and 14.65 Ω, indicating the carbon coating leads to a more stable SEI which is beneficial to the electron transfer. In addition, the slope of the TNO@C-2 electrode is steeper than that of the TNO electrode, standing for a quicker Li^+^ diffusion. The EIS results are consistent with superior rate performance of TNO@C-2. Using a four-point probing approach, the electrical conductivity of the TNO and TNO@C-2 powders was investigated in order to better understand how carbon-coating contributes to the improvement of electrochemical performance. TNO@C-2 has a much better electron conductivity (3.16 × 10^−6^ S cm^−1^), which is higher than that of TNO (8.91× 10^−12^ S cm^−1^). The results show that the carbon coating significantly improved the electronic conductivity and decreased the charge transfer resistance of TNO@C-2 [29].

As shown in Figure 6b,c, we also recorded CV curves at different scan rates from 0.1 to 1 mV s^−1^ to analyze the charge storage mechanisms and reasons for excellent rate capability. The CV curves exhibit identical redox pairs between 1.0 and 3 V, with the redox peaks increasingly deviating from the conventional position. Peak currents (*i*) are often exponentially connected to sweeping rates (*v*), as illustrated below [30]:*i* = *av**^b^*(1)
where *i* is the peak current of CV, *v* is the scan rate, and *a* and *b* are adjustable parameters. The charge storage mechanism can be determined from the *b* values, obtained from the log(*v*) plotted as a function of log(*i*) (Figure 6d,e). A dominant diffusion-controlled behavior or capacitive electrochemical performance is predicted by *b* = 0.5 or 1.0 [31]. The b values exhibited by the TNO and TNO@C-2 electrodes were equal to 0.51/0.69 and 0.67/0.86 for the cathodic and anodic processes, respectively. Moreover, higher b-values for the carbon coating active material confirm the results discussed above, that theTNO@C-2 could be characterized by a dominant capacitive behavior and fast kinetics. To understand the hybrid behavior mechanisms, we accessed the quantitative capacitive contributions for TNO and TNO@C-2electrodes as shown below [32]:*i(v)* = *k*_1_*v* + *k*_2_*v*^1/2^(2)
where *k*_1_*v* and *k*_2_*v*^1/2^ are contributions to the surface and diffusion-controlled behaviors, respectively. The capacitance contribution of TNO@C-2 was higher than that of TNO. Additionally, the capacitive contribution of the TNO@C-2 was higher at all scan rates (Figure 6f). Thus, carbon coating accelerates Li^+^ transport and improves the rate performance and long-cycling stability of the active material.

## 4. Conclusions

In conclusion, the self-polymerization coating properties of dopamine were used to form functionalized nano-coatings to modify TNO anodes. The carbon layer is uniformly coated on the surface of the TNO particles. Compared with the pristine TNO particles, the electronic conductivity of TNO@C-2 has been significantly improved, resulting in the excellent rate performance and cycle stability. Furthermore, carbon coating is a common method to improve electrical conductivity. At 0.1 C, TNO@C-2 delivered 295.6 mA h g^−1^ capacity with coulombic efficiency equal to 99.2%. The TNO@C-2 based cell also exhibited excellent durability, maintaining a capacity of 164.1 mA h g^−1^ after 400 cycles at 10 C. In addition, the EIS results demonstrate the carbon coating effectively improves the rate performance. We believe that our results and data will help to develop new strategies to improve the ionic/electronic conductivity of other electrode materials.

## Figures and Tables

**Figure 1 materials-16-00333-f001:**
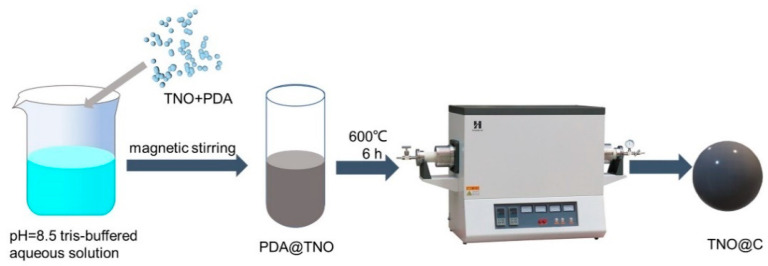
Schematic diagram of the preparation of TNO@C.

**Figure 2 materials-16-00333-f002:**
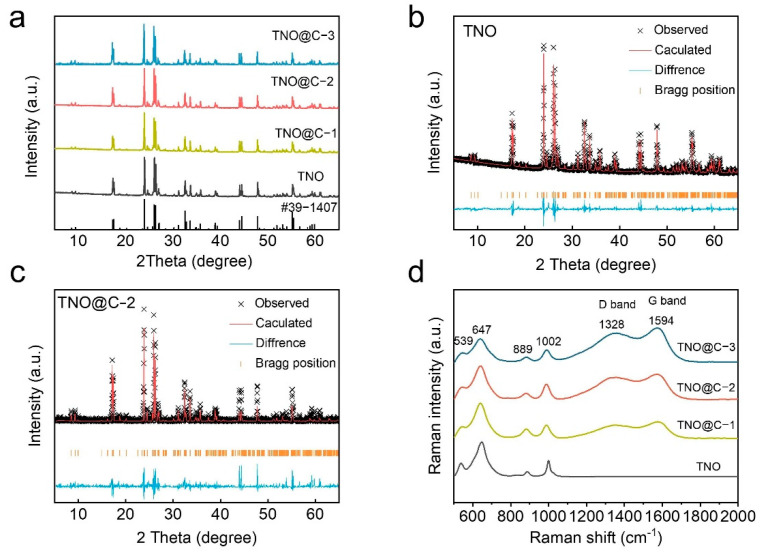
(**a**) XRD patterns of TNO and TNO@C; Refinement patterns of XRD for (**b**) TNO and (**c**) TNO@C-2 material; (**d**) Raman spectrum of TNO and TNO@C.

**Figure 3 materials-16-00333-f003:**
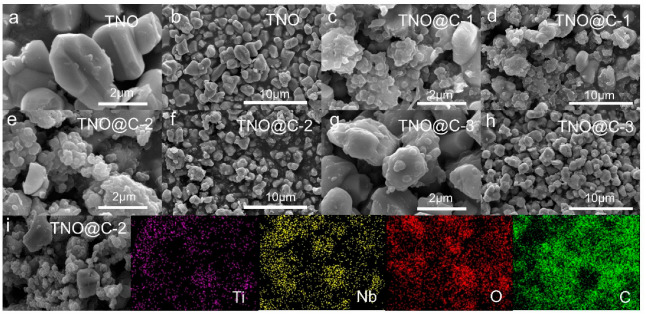
SEM images of (**a**,**b**) TNO, (**c**,**d**) TNO@C-1, (**e**,**f**) TNO@C-2 and (**g**,**h**) TNO@C-3. (**i**) EDS mapping of Ti, Nb, O, and C elements.

**Figure 4 materials-16-00333-f004:**
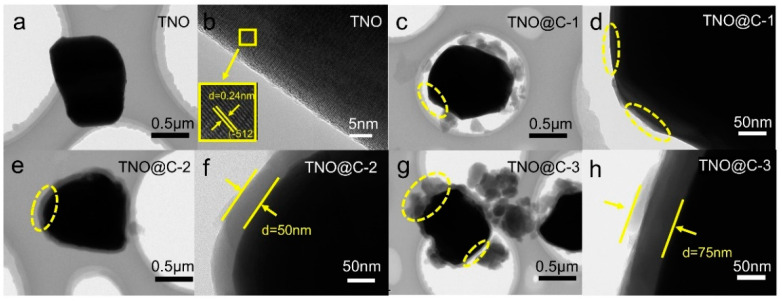
TEM images of (**a**,**b**) TNO (**c**,**d**) TNO@C-1, (**e**,**f**) TNO@C-2 and (**g**,**h**) TNO@C-3.

**Figure 5 materials-16-00333-f005:**
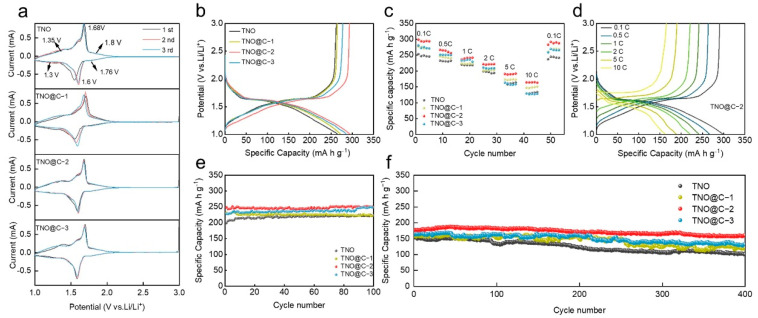
(**a**) CV profiles of TNO, TNO@C at 0.1 mV s^−1^; (**b**) Discharge/charge profiles of TNO and TNO@C in the FIRST cycles at 0.1 C; (**c**) rate performance of TNO and TNO@C; (**d**) discharge/charge curve of TNO@C-2 at different C-rates; cycling stability of TNO and TNO@C-2 at (**e**) 1 C and (**f**) 10 C.

**Figure 6 materials-16-00333-f006:**
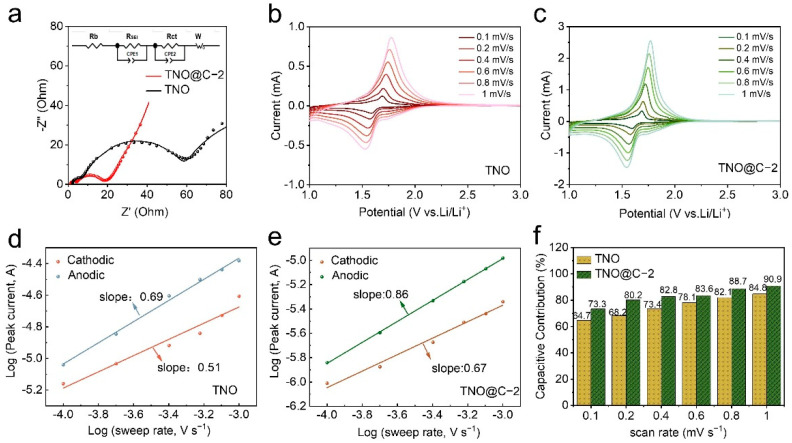
(**a**) Nyquist plots of TNO and TNO@C-2; (**b**) electrical conductivity of the TNO and TNO@C-2; CV curves of (**c**) TNO and (**d**) TNO@C-2 at various scan rates; (**e**) the corresponding log (*i*) versus log (*v*) plots of TNO and TNO@C-2; (**f**) capacitive contribution ratios at different scan rates.

**Table 1 materials-16-00333-t001:** The summarization of TNO anode materials and their capacities.

Material	Reversible Specific Capacity (mA h g^−1^)	High-Rate Capability (mA h g^−1^)	Mass Loading (mg^.^cm^−2^)
Porous TNO@C [14]	~280 at 0.1 C	211 at 10 C	2.3
Ti_2_Nb_10_O_29_/C microspheres [15]	275 at 1 C	214 at 30 C	1.5
TNO@C [16]	265 at 0.5 A g^−1^	75 at 6 A g^−1^	1.5–2.0
NPTNO MS-3 [17]	265 at 0.1 C	~120 at 30 C	---
TiNb_2_O_7_/CNT-KB [18]	327 at 0.1 C	151 at 20 C	1.0
TNO/CNT [21]	346 at 0.1 C	163 at 30 C	1.4
Hollow TNO@C spheres [22]	283 at 0.25 C	157 at 10 C	---

## Data Availability

Not applicable.

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
