# Peer review of "Establish TiNb2O7@C as Fast-Charging Anode for Lithium-Ion Batteries"

_materials, 2022, doi:10.3390/ma16010333_

Round 1
Reviewer 1 Report
To the authors:
The authors present a comparative study on TiNb2O7 without and with different carbon coatings. While the application of a carbon coating is a common step and not particularly novel, the results are promising. However, I suggest the authors to follow the comments below and add all the relevant experiments and information in order to improve the quality of the manuscript and to further solidify the drawn conclusions.
General remarks
Comment #1: The manuscript is well written in general. However, the English has to be revised again before publication.
Comment #2: I suggest the authors provide a table comparing the present electrochemical results with the most important works (described in the introduction). This is very helpful for the reader in order to clearly judge the performance. This table should include the active material mass loading, which is commonly very low for studies reporting high power materials.
Experimental section:
Comment #3: Which kind of furnaces were used for the synthesis of TNO (2.1) and the carbon coatings (2.2)?
Comment #4: Please add the active material mass loading of the electrodes as well as the electrode thickness (including and excluding the Cu foil) to the experimental section. This is essential information, for instance to really judge the electrochemical performance, or to get an idea of the electrode porosity.
Comment #5: Please be more specific in describing the cells that were used: What was the counter electrode? If it was lithium then provide the supplied and purity (and thickness). What was the separator, how thick was it and how much electrolyte was used for that separator.
Results and Discussion:
Comment #6: The authors claim that there are no obvious phase impurities present in the diffractograms in Figure 2. However, I think there is a substantial amount of reflections which do not match with the reference (e.g. at 20° right after the first large reflection). These impurities should be identified and labeled.
Comment #7: “Furthermore, the samples of 115 TNO@C exhibit the same diffraction peaks as TNO, proving that the TNO particles were 116 not reduced (Fig.2a).”
In fact, some of the impurities seem to disappear with increasing carbon content. Is it possible that these are reduced?
Comment #8: The presence of carbon in the samples is nicely demonstrated. However, the authors should determine the amount of carbon in each of the three carbon coated materials, for example via simple TGA measurements.
Comment #9: “The position of the CV peak at the cathode in the first…”
The CV peaks appear during “cathodic” and “anodic sweeps”. Declaring the electrode as a cathode is rather misleading and does not follow the state-of-the-art in battery publications. I suggest to call the electrode anode or negative electrode, even though the tests may be performed in half cells with lithium metal. This applies for all paragraphs on CVs.
Comment #10: Figure5: It should be clearly indicated in the text to which mass the specific capacity is referred to. If it is referred to the active material including the carbon then the authors should provide another panel with performance data where the specific capacity is referenced solely to the active material (NTO) excluding the amount of carbon. Again, the active material mass loading should be provided in the experimental section and the amount of carbon should be determined via TGA (for instance).
Comment #11: The impedance related to SEI (RSEI) describes how fast lithium ions diffuse through that particular solid electrolyte interphase. This is a bit misleading in the manuscript, describing it as “insertion into the particle surface”. I would assume that the carbon coating leads to a more stable SEI, which would have to be proven by XPS measurements.
Comment #12: Is there a reason for the peaks not following a clear trend in Figure 6c for the TNO@C-2?
Author Response
Responses to Reviewers’ comments
Dear Editor:
Thank you and the reviewers for the positive and constructive comments about our manuscript entitled “Establish TiNb2O7@C as fast-charging anode for lithium-ion batteries” (ID: materials-2005671). According to your nice suggestion, we have made extensive corrections to our pervious draft, the detailed corrections are listed below.
Reviewer: 1
Comment #1: The manuscript is well written in general. However, the English has to be revised again before publication.
Response: Thank you for the comment. We tried our best to improve the manuscript and made some changes to the manuscript. These changes will not influence the content and framework of this paper. We hope the revised manuscript could be acceptable for you.
Comment #2: I suggest the authors provide a table comparing the present electrochemical results with the most important works (described in the introduction). This is very helpful for the reader in order to clearly judge the performance. This table should include the active material mass loading, which is commonly very low for studies reporting high power materials.
Response: Thank you for the comment and the supplement elucidation presents in the introduction section as “Recently, various methods have been used to synthesize TNO anode materials and the electrochemical properties of the various TNO anodes have been supplemented in the supporting information (Table 1)”
Table 1 The summarization of TNO anode materials and their capacities at low and high rates in recently published work.
|
Material |
Reversible specific capacity |
High-rate capability |
mass loading (mg.cm-2) |
|
Porous TNO@C [14] |
~280mA h g-1 at 0.1 C |
211 mA h g-1 at 10C |
2.3 |
|
Ti2Nb10O29/C microspheres [15] |
275mA h g-1 at 1 C |
214 mA h g-1 at 30 C |
1.5 |
|
TNO@C[16] |
265 mA h g-1 at 0.5 A g-1 |
75 mA h g-1 at 6 A g-1 |
1.5-2.0 |
|
NPTNO MS-3 [17] |
265mA h g-1 at 0.1 C |
~120 mA h g-1 at 30 C |
--- |
|
TiNb2O7/CNT-KB [18] |
327 mA h g-1 at 0.1 C |
151 mA h g-1 at 20 C |
1.0 |
|
TNO/CNT[23] |
346 mA h g-1 at 0.1 C |
163 mA h g-1 at 30 C |
1.4 |
|
Hollow TNO@C spheres[24] |
283 mA h g-1 at 0.25 C |
157 mA h g-1 at 10 C |
--- |
Experimental section:
Comment #3: Which kind of furnaces were used for the synthesis of TNO (2.1) and the carbon coatings (2.2)?
Response: Thank you for the comment. The type of furnace is GSL-1400. We have supplemented in the manuscript and marked.
Comment #4: Please add the active material mass loading of the electrodes as well as the electrode thickness (including and excluding the Cu foil) to the experimental section. This is essential information, for instance to really judge the electrochemical performance, or to get an idea of the electrode porosity.
Response: Thank you for the comment. The supplement elucidation presents as: “The loading mass of the active material in the prepared electrodes was controlled in the range of 1.0~1.5 mg cm-2. The electrode thickness was about 25-30 μm and the Cu foil is 12 μm.”
Comment #5: Please be more specific in describing the cells that were used: What was the counter electrode? If it was lithium then provide the supplied and purity (and thickness). What was the separator, how thick was it and how much electrolyte was used for that separator.
Response: Thank you for the comment. The supplement elucidation presents as: “Lithium mental (China Energy Lithium Co., Ltd., diameter × thickness: Ï•16 × 2 mm, >99.9%) was used as the counter electrode, and the separator consisted of a polypropylene membrane (Celgard H1609, thickness:16μm) that had been moistened with about 100µL electrolyte (1 M LiPF6 and 1:1 volume combination of ethylene carbonate (EC) and Diethyl carbonate (DEC)).”
Results and Discussion:
Comment #6: The authors claim that there are no obvious phase impurities present in the diffractograms in Figure 2. However, I think there is a substantial amount of reflections which do not match with the reference (e.g. at 20° right after the first large reflection). These impurities should be identified and labeled.
Response: Thank you for the comment. In fact, the peak at 20 degrees on the PDF card is not shown in the image due to an inappropriate range of vertical coordinates, which we have corrected.
Figure 2. (a) XRD patterns of TNO and TNO@C; Refinement patterns of XRD for (b) TNO and (c) TNO@C-2 material; (d) Raman spectrum of TNO and TNO@C.
Comment #7: “Furthermore, the samples of 115 TNO@C exhibit the same diffraction peaks as TNO, proving that the TNO particles were 116 not reduced (Fig.2a).” In fact, some of the impurities seem to disappear with increasing carbon content. Is it possible that these are reduced?
Response: Thank you for the comment. The XRD results were compared with the PDF card (#77-1374) and found that all the characteristic peaks matched. The problem you mentioned was due to the inappropriate range of vertical coordinates, and the corrected image is below.
Figure 2. (a) XRD patterns of TNO and TNO@C; Refinement patterns of XRD for (b) TNO and (c) TNO@C-2 material; (d) Raman spectrum of TNO and TNO@C.
Comment #8: The presence of carbon in the samples is nicely demonstrated. However, the authors should determine the amount of carbon in each of the three carbon coated materials, for example via simple TGA measurements.
Response: Thank you for the comment. The supplement elucidation presents as: “The amount of carbon in the TNO@C samples was tested by organic elemental analysis and the results showed that the amount of carbon in the samples were about 4.7, 5.2, and 6.4%.”
Comment #9: “The position of the CV peak at the cathode in the first…”
The CV peaks appear during “cathodic” and “anodic sweeps”. Declaring the electrode as a cathode is rather misleading and does not follow the state-of-the-art in battery publications. I suggest to call the electrode anode or negative electrode, even though the tests may be performed in half cells with lithium metal. This applies for all paragraphs on CVs.
Response: Thank you for the comment. All statements in the manuscript on CVs have been corrected into electrode anode/cathode.
Comment #10: Figure5: It should be clearly indicated in the text to which mass the specific capacity is referred to. If it is referred to the active material including the carbon then the authors should provide another panel with performance data where the specific capacity is referenced solely to the active material (NTO) excluding the amount of carbon. Again, the active material mass loading should be provided in the experimental section and the amount of carbon should be determined via TGA (for instance).
Response: Thank you for the comment. We have supplied the active material mass loading and the amount of carbon in the experimental section. The amount of carbon is less and we consider TNO@C as a whole to be the active materials due to TNO@C was obtain via calcination method, the TNO and carbon are compounded together.
Comment #11: The impedance related to SEI (RSEI) describes how fast lithium ions diffuse through that particular solid electrolyte interphase. This is a bit misleading in the manuscript, describing it as “insertion into the particle surface”. I would assume that the carbon coating leads to a more stable SEI, which would have to be proven by XPS measurements.
Response: Thank you for the comment. The expression of the “insertion into the particle surface” is inaccurate, and it has been modified: “On the other hand, the TNO@C-2 are just 2.6 and 14.65 Ω, indicating the carbon coating leads to a more stable SEI which is beneficial to the electron transfer.”
Comment #12: Is there a reason for the peaks not following a clear trend in Figure 6c for the TNO@C-2?
Response: Thank you for the comment. The peak assignment was inaccurate and the corrected image is below.
Figure 6 (a) Nyquist plots of TNO and TNO@C-2; (b) electrical conductivity of the TNO and TNO@C-2; CV curves of (c) TNO and (d) TNO@C-2 at various scan rates; (e) the corresponding log (i) versus log (v) plots of TNO and TNO@C-2; (f) capacitive contribution ratios at different scan rates.

Reviewer 2 Report
Refer to the attached document for specific comments.

Author Response
Responses to Reviewers’ comments
Dear Editor:
Thank you and the reviewers for the positive and constructive comments about our manuscript entitled “Establish TiNb2O7@C as fast-charging anode for lithium-ion batteries” (ID: materials-2005671). According to your nice suggestion, we have made extensive corrections to our pervious draft, the detailed corrections are listed below.
Reviewer: 2
Abstract
No comment
Introduction
(Page 1, line 31 and others) check the proper way to write the citation.
Response: Thank you for the comment. We have checked the way to write citation and make the corrected.
(Page 1, line 37-38) add information to bridge between the safety issue of graphite to Niobium-
titanium oxide and why chose it
Response: Thank you for the comment. The supplement elucidation presents as: “Due to the decreased risk of Li plating at low voltages, metal oxides with intercalation-type lithium storage mechanisms are being considered as an alternative anode material for safe LIBs. Niobium-titanium oxides are the intercalation-type anode materials which could provide fast charging capability for lithium-ion batteries and have excellent structural stability during lithiation/delithiation.”
(Page 1, line 30-40) please elaborate on the importance of this information!
Response: Thank you for the comment. We have made some changes in the manuscript to make the information clearer.
(Page 1, line 40-41) bridge why before statement has relation to all beneficial factors, especially
safety, which is used to compare with graphite.
Response: Thank you for the comment. We state at 47-49 that: “Furthermore, owing to the lithiation voltage (1.6 V vs. Li+/Li), Li dendrite and SEI film production are avoided during the charging/discharging process, ensuring the safety of the battery.”
(Page 1, line 41-34) add information on graphite theoretical value as a comparison.
Response: Thank you for the comment. The supplement elucidation in 32-34 presents as: “With the advantage of inexpensive, structural stability and high specific capacity (372 mA h g-1), graphite materials are commonly employed in LIBS, but the production of SEI films during the first cycle causes a partial loss of capacity and is prone to the precipitation of lithium dendrites, which leads to internal short circuits in the battery, posing a severe safety threat.”
Materials & Methods
(Page 2, line 72) complete the information, e.g., "TiO2 from Sigma-Aldrich (Saint Louis, MO,
USA), with a purity of 99%, … and …."
Response: Thank you for the comment. We have completed the information: “TNO was made by a one-step solid state reaction using TiO2 from Macklin (Shanghai, China), with a purity of 99.8%, and Nb2O5 Macklin (Shanghai, China), with a purity of 99.9%, …”
(Page 2, line 78) change 2:1/1:1/1:2 to 2:1, 1:1 and 1:2.
Response: Thank you for the comment. We have corrected it.
(Page 2, line 81-82) move to the beginning of the paragraph
Response: Thank you for the comment. We have corrected it.
(Page 2, line 85) complete the information of the maker's address, such as, e.g., "(Tokyo, Japan)" and do to others as well
Response: Thank you for the comment. We have completed the information.
(Page 2, line 85-86) complete the information on instrument configuration (optics), kV-mA, 2theta range, step size and dwell time. Sample preparation is also essential for XRD measurement.
Response: Thank you for the comment. We have completed the information: “The resulting materials were analyzed by powder X-ray diffraction (XRD) performed by a Rigaku-D/MAX-rA instrument (Netherlands) equipped with a Cu-Ka radiation source under 40 mA and 40 kV. The XRD patterns were collected over the 2θ range of 5–65â—¦ with a step size of 0.02â—¦, and the scanning rate was 2â—¦ min−1.”
(Page 2, line 86-92) complete the information on each instrument above
Response: Thank you for the comment. We have completed the information.
(Page 3, line 98) write the extended version of EC and DEC before using their abbreviation
Response: Thank you for the comment. We have completed the extended version of EC and DEC: “…ethylene carbonate (EC) and Diethyl carbonate (DEC).”
(Page 3, line 99-101) complete the information on the maker's address
Response: Thank you for the comment. We have completed the information.
Result & Discussion
(Page 3, line 105-110) move to the material & method section
Response: Thank you for the comment. We have moved it to the material & method section.
(Page 3, line111-112) Do the Rietveld refinement for the detail of the crystal structure.
Response: Thank you for the comment. We have made the Rietveld refinement for the detail of the crystal structure and the results are presented in Fig.2b-c
Figure 2. (a) XRD patterns of TNO and TNO@C; Refinement patterns of XRD for (b) TNO and (c) TNO@C-2 material; (d) Raman spectrum of TNO and TNO@C.
(Page 3, line 111-114) JCPDS has not existed since 1978. It is now known as ICDD, and the card no will be AA-BBB-XXXX; where AA means "00" – ICDD; "01" – ICSD; "02" –CSD; "03" – NIST; "04" – LPF; and "05" – ICDD, BBB, corresponding to the annual publication, and a pattern number (XXXX).
Response: Thank you for the comment. We denote it as: 00-434-3509
(Page 3, line 114-115) the scan does not start at a low enough angle, and it misses the first three
essential peaks (the first peak @8.5 deg)
Response: Thank you for the comment. The scan started at 5 degree and the corrected image is below.
Figure 2. (a) XRD patterns of TNO and TNO@C; Refinement patterns of XRD for (b) TNO and (c) TNO@C-2 material; (d) Raman spectrum of TNO and TNO@C.
(Page 3, line 115-117) There is no way to detect the particle reduction since it was polycrystalline. The one that can be reduced is the crystallite size which is indicated by the peak broadening.
Response: Thank you for the comment. In this sentence “Furthermore, the samples of TNO@C exhibit the same diffraction peaks as TNO, proving that the TNO particles were not reduced (Fig.2a).” What we want to convey is that TNO does not undergo a redox reaction with carbon, not that the TNO particle size becomes smaller.
(Page 4, line 129) supposed to be Fig.2c-h and Fig.2i
Response: Thank you for the comment. We feel sorry for our carelessness. In our manuscript, the mistake is revised. Thanks for your corrections.
(Figure 3i,) why is no/less carbon detected in 2 big particles? Try to improve the image quality.
Response: Thank you for the comment. The small particles in the picture are the carbon, which is also confirmed by the Mapping of C.
(Page 4, line 136-117) if the crystallographic setting is A2/m then (21-5) corresponds to d=0.276 nm, if the setting is I2/m then (-512) corresponds to d=0.243 nm.
Response: Thank you for the comment. We feel sorry for our carelessness. In our manuscript, the mistake is revised. Thanks for your corrections.
Figure.4 TEM images of (a-b) TNO (c-d) TNO@C-1, (e-f) TNO@C-2 and (g-h) TNO@C-3.
(Page 4, line 147-149) please move to the materials & methods section or everything related to
the test procedure and instrument settings. Check also the 1C and 10C
Response: Thank you for the comment. We have corrected it.
(Page 5, line 153-156) marks the values on the pictures
Response: Thank you for the comment. We have marked the values on the pictures.
Figure 5. (a) CV profiles of TNO, TNO@C at 0.1mV s-1; (b) Discharge/charge profiles of TNO and TNO@C in the FIRST cycles at 0.1 C; (c) rate performance of TNO and TNO@C; (d) dis-charge/charge curve of TNO@C-2 at different C-rates; cycling stability of TNO and TNO@C-2 at (e) 1 C and (f) 10 C.
(Page 5, line 170-180) Rewrite for clarity by stating the 100 and then 400 cycles to further demonstrate the effect on carbon coating.
Response: Thank you for the comment. We have rewritten it: “The TNO@C-2 anode showed that the discharge capacity of 248.7 mA h g-1 after 100 cycles at 1C (Fig. 5e). Furthermore, TNO@C-2 exhibited the best electrochemical performance with the highest capacity and excellent cycling stability. Even at 10C, TNO@C-2 displayed an initial discharge capacity of 177.4 mA h g-1 and the capacity only reduced by roughly 9% after 400 cycles, the capacity retention of TNO@C-2 is higher than TNO@C-1 (78.8%) and TNO@C-3 (79.6%). While, TNO only provided the discharge capacity of 101.3 mA h g-1 with 65 % retention after 400 cycles at 10 C. (Fig.5f)”
(Page 5, line 183) rewrite for clarity: "discharge/charge curve of TNO@C-2 at…"
Response: Thank you for the comment. We have rewritten it: “discharge/charge curve of TNO@C-2 at different C-rates.”
(Page 5, line 184) rewrite for clarity: "cycling stability of TNO and TNO@C-2 at…"
Response: Thank you for the comment. We have rewritten it: “cycling stability of TNO and TNO@C-2 at (e) 1 C and (f) 10 C.”
(Page 5, line 187-189) state the approximate range and region
Response: Thank you for the comment. The supplement elucidation presents as: “Fig. 6a depicts the appropriate Nyquist plots and equivalent circuit, which consists of a semicircle in the high and medium frequency ranges (106-104, 104-101) and a sloping line in the low frequency region (101-10-2).”
Conclusion
(Page 7, line 233-234) rewrite for clarity
Response: Thank you for the comment. The supplement elucidation presents as: “The carbon layer is uniformly coated on the surface of the TNO particles. Compared with the pristine TNO particles, the electronic conductivity of TNO@C-2 has been significantly improved, resulting in the excellent rate performance and cycle stability. Furthermore, carbon coating is a common method to improve electrical conductivity.”
(Page 7, line 239-241) consider deleting this and adding the conclusion from EIS
Response: Thank you for the comment. The supplement elucidation presents as: “In addition, the EIS results demonstrate the carbon coating effectively improves the rate performance.”
References
Add DOI to all references
Response: Thank you for the comment. We have added DOI to all references.

Round 2
Reviewer 1 Report
The authors have nicely addressed all my comments and I believe the manuscript can be published in its current state.
Thank you.